# *MIA3* Splice Defect in Cane Corso Dogs with Dental-Skeletal-Retinal Anomaly (DSRA)

**DOI:** 10.3390/genes12101497

**Published:** 2021-09-25

**Authors:** Matthias Christen, Henriëtte Booij-Vrieling, Jelena Oksa-Minalto, Cynthia de Vries, Alexandra Kehl, Vidhya Jagannathan, Tosso Leeb

**Affiliations:** 1Institute of Genetics, Vetsuisse Faculty, University of Bern, 3001 Bern, Switzerland; matthias.christen@vetsuisse.unibe.ch (M.C.); vidhya.jagannathan@vetsuisse.unibe.ch (V.J.); 2Department of Clinical Sciences of Companion Animals, General Surgery, Faculty of Veterinary Medicine, Utrecht University, 3584 CM Utrecht, The Netherlands; h.e.booij-vrieling@uu.nl; 3Kengaraga Veterinary Clinic, LV-1035 Riga, Latvia; vet.dr.oksa@gmail.com; 4Laboklin GmbH & Co. KG, Steubenstraße 4, 97688 Bad Kissingen, Germany; devries@laboklin.com (C.d.V.); kehl@laboklin.com (A.K.)

**Keywords:** *Canis lupus familiaris*, animal model, endoplasmic reticulum, TANGO1, collagen, precision medicine, non-coding, splicing

## Abstract

We investigated a hereditary syndrome in Cane Corso dogs. Affected dogs developed dental-skeletal-retinal anomaly (DSRA), clinically characterized by brittle, discolored, translucent teeth, disproportionate growth and progressive retinal degeneration resulting in vision loss. Combined linkage and homozygosity mapping delineated a 5.8 Mb critical interval. The comparison of whole genome sequence data of an affected dog to 789 control genomes revealed a private homozygous splice region variant in the critical interval. It affected the *MIA3* gene encoding the MIA SH3 domain ER export factor 3, which has an essential role in the export of collagen and other secreted proteins. The identified variant, XM_005640835.3:c.3822+3_3822+4del, leads to skipping of two exons from the wild type transcript, XM_005640835.3:r.3712_3822del. Genotypes at the variant were consistent with monogenic autosomal recessive mode of inheritance in a complete family and showed perfect genotype-phenotype association in 18 affected and 22 unaffected Cane Corso dogs. *MIA3* variants had previously been shown to cause related phenotypes in humans and mice. Our data in dogs together with the existing functional knowledge of *MIA3* variants in other mammalian species suggest the *MIA3* splice defect and a near complete loss of gene function as causative molecular pathomechanism for the DSRA phenotype in the investigated dogs.

## 1. Introduction

Skeletal dysplasias are a heterogeneous and large group of inherited disorders associated with abnormalities in development, growth and homeostasis of the skeleton. They frequently result in short stature [1]. To date, the Nosology Committee of the International Skeletal Dysplasia Society recognizes 42 diverse groups of skeletal dysplasias in humans, with causative variants identified in over 400 different genes [2]. Many of the described diseases are not restricted to a skeletal phenotype, but manifest as syndromic conditions that additionally involve other tissues and organ systems.

In dogs, a syndromic skeletal dysplasia with known genetic cause is the oculoskeletal dysplasia (OSD). This syndrome is a combination of retarded growth leading to short-limbed dwarfism, together with ocular signs such as cataract or retinal detachments and was first described over 40 years ago [3,4,5]. Today, disease-causing variants for OSD have been identified in *COL9A2* for Samoyed dogs and in *COL9A3* for Labrador Retrievers and Northern Inuit dogs (OMIA 001522-9615, 001523-9615) [6,7].

In the present study, we investigated a cohort of Cane Corso dogs suffering from a new syndromic phenotype that we tentatively termed dental-skeletal-retinal anomaly (DSRA). The goal of the study was to investigate a possible underlying causative genetic defect.

## 2. Materials and Methods

### 2.1. Clinical and Pathological Examinations

The study comprised 18 DSRA affected and 22 control Cane Corso dogs. A comprehensive physical examination including macroscopical inspection of the oral cavity was performed on all DSRA affected dogs. Ophthalmological examinations were performed on selected DSRA affected dogs. The examination consisted of a slit lamp biomicroscopy (KJ5S3, Suzhou Kangjie Medical, Suzhou, China), tonometry (iCareTonovet Plus, Icare, Vantaa, Finland), indirect ophthalmoscopy (Heine omega 500, Gilching, Germany), chromatic pupil light reflex testing (red light wavelength 630 nm/blue light wavelength 457 nm), Schirmer’s tear test and fluorescein eye stain test. Fundus images were obtained with a retinal camera (Optibrand ClearView, Fort Collins, CO, USA).

EDTA blood samples were collected for genomic DNA isolation. From one DSRA affected dog, additional muscle tissue samples were taken within two hours after euthanasia and preserved in RNAlater (Thermo Fisher Scientific, Waltham, MA, USA) for the isolation of total RNA.

### 2.2. DNA and SNV Genotyping

Genomic DNA was isolated from EDTA blood samples with the Maxwell RSC Whole Blood DNA Kit using a Maxwell RSC instrument (Promega, Dübendorf, Switzerland). DNA from 12 affected and 8 unaffected animals was genotyped on illumina_HD canine BeadChips containing 220,853 markers (Neogen, Lincoln, NE, USA). The raw SNV genotypes are available in Appendix A. We did not have complete pedigree information on all 20 dogs that were genotyped on the SNV arrays. Some of the dogs were closely related, including, for example, one complete family with two affected full siblings and one healthy puppy, which was used for parametric linkage analysis.

### 2.3. Linkage Analysis and Homozygosity Mapping

For linkage analysis, we worked with one family consisting of five dogs. For all dogs, the call rate was >95%. Using PLINK v1.9 [8], markers that were located on the sex chromosomes or missing in any of the dogs, had Mendel errors or a minor allele frequency < 0.01, were removed. The final pruned dataset contained 103,168 markers. To analyze the data for parametric linkage, an autosomal recessive inheritance model with full penetrance, a disease allele frequency of 0.5 and the Merlin software [9] were applied.

For homozygosity mapping, the genotype data from 12 affected dogs were used. Markers that were missing in one of the 12 cases and markers on the sex chromosomes were excluded. The PLINK options --homozyg-group, --homozyg-kb 200, --homozyg-snp 30 and --homozyg-window-snp 30 were used for the analysis. The output intervals were intersected with the intervals from the linkage analysis in an Excel spreadsheet to find overlapping regions (Appendix A). A tped-file containing the markers on chromosome 38 was visually inspected in an Excel spreadsheet to double check the homozygous shared haplotype in the cases (Appendix A). All positions correspond to the CanFam3.1 reference genome assembly.

### 2.4. Whole-Genome Sequencing

An Illumina TruSeq PCR-free DNA library with ~400 bp insert size of an affected dog (CI009) was prepared. We collected 252 million 2 × 150 bp paired-end reads on a NovaSeq 6000 instrument (29.0× coverage). Mapping and alignment were performed as described [10]. The sequence data were deposited under the study accession PRJEB16012 and the sample accession SAMEA8157164 at the European Nucleotide Archive.

### 2.5. Variant Calling

Variant calling was performed using GATK HaplotypeCaller [11] in gVCF mode as described [10]. To predict the functional effects of the called variants, SnpEff [12] software together with NCBI annotation release 105 for the CanFam3.1 genome reference assembly was used. For variant filtering, we used 789 control genomes from wolves and dogs of diverse breeds (Appendix A).

### 2.6. Gene Analysis

We used the CanFam3.1 dog reference genome assembly and NCBI annotation release 105. Numbering within the canine *MIA3* gene corresponds to the NCBI RefSeq accession numbers XM_005640835.3 (mRNA) and XP_005640892.1 (protein).

### 2.7. Sanger Sequencing

The *MIA3*:c.3822+3_3822+4delTA variant was genotyped by direct Sanger sequencing of PCR amplicons. A 359 bp (or 357 bp in case of the mutant allele) PCR product was amplified from genomic DNA using AmpliTaqGold360Mastermix (Thermo Fisher Scientific) together with primers 5′-TAT GGA TTT CCC CTC CCT TT-3′ (Primer F) and 5′-AAC CAC AGG GCT ATC AGA ACT T-3′ (Primer R). After an initial denaturation of 10 min at 95 °C, 30 cycles of 30 s at 95 °C, 30 s at 60 °C, and 60 s at 72 °C were performed, followed by a final extension step of 7 min at 72 °C. PCR products were treated with exonuclease I and alkaline phosphatase. Subsequently, the amplicons were sequenced on an ABI 3730 DNA Analyzer (Thermo Fisher Scientific). Sanger sequences were analyzed using the Sequencher 5.1 software (GeneCodes, Ann Arbor, MI, USA).

### 2.8. RNA Isolation and RT-PCR

Total RNA from skeletal muscle tissue of a DSRA affected dog was extracted using the RNeasy Mini Kit (Qiagen, Hilden, Germany). The RNA was cleared of genomic DNA contamination using the QuantiTect Reverse Transcription Kit (Qiagen). The same kit was used to synthetize cDNA, as described by the manufacturer. For RT-PCR, a forward primer 5′-CCT TCT TGG GAA TTG GTT CA-3′ located in exon 7 together with a reverse primer 5′-AGC TGT ATC GTC CAG AAT TTC A-3′ located in exon 11 were used. A control cDNA derived from skin of a non-affected dog was obtained during a previous study of our group [13]. After an initial denaturation of 10 min at 95 °C, 35 cycles of 30 s at 95 °C, 30 s at 60 °C, and 60 s at 72 °C were performed, followed by a final extension step of 7 min at 72 °C. The RT-PCR products were visualized using a 5200 Fragment Analyzer instrument (Agilent, Basel, Switzerland), and sequenced as described above.

## 3. Results

### 3.1. Clinical Description

Eighteen Cane Corso dogs presented with a similar combination of clinical signs affecting their teeth, skeletal morphology and vision. They had brittle and translucent deciduous and permanent teeth showing marked brown/pink discoloration and multifocal enamel defects (Figure 1). The affected dogs were small in size in comparison with their littermates and displayed disproportionate growth with short and bent legs.

Affected Cane Corso dogs developed signs of vision loss. Indirect ophthalmoscopy revealed bilateral retinal changes that could be classified as progressive retinal atrophy (PRA; Figure 2).

### 3.2. Genetic Analysis

The occurrence of DSRA in multiple puppies of a litter with healthy parents suggested autosomal recessive inheritance. Parametric linkage analysis in a family consisting of the parents and three offspring identified 50 linked segments spanning 446 Mb with a maximum LOD score of 0.73. Homozygosity mapping in 12 DSRA affected dogs identified 26 extended homozygous regions with shared haplotypes. A total of 10 genomic segments on five different chromosomes showed simultaneous linkage in the family with a maximum LOD score of 0.73 and homozygosity in the 12 cases. Taken together, these 10 intervals spanned 5.8 Mb or roughly 0.24% of the 2.4 Gb dog genome and were considered the critical interval for the subsequent analyses (Appendix A).

The genome of one affected dog was sequenced and searched for homozygous private variants by comparing the variants from the case with 789 control genomes (Table 1 and Appendix A).

The analysis identified a single private homozygous variant with likely functional impact on a protein in the critical interval. This variant, an intronic 2 bp deletion, was located in the *MIA3* gene. It can be designated as Chr38:16,920,529_16,920,530delAT (CanFam3.1 assembly) or XM_005640835.3:c.3822+3_3822+4del. With respect to the annotated *MIA3* transcript isoform XM_005640835.3, the 2 bp deletion was located in the 5′-splice site of intron 9 (Figure 3A). The presence of the deletion was confirmed by Sanger sequencing and the available 40 Cane Corso dogs were genotyped. All DSRA cases and none of the control dogs were homozygous for the variant (Table 2). The genotypes also showed the expected co-segregation in the family (Figure 3B).

### 3.3. Functional Confirmation at the Transcript Level

As the genomic variant did not directly affect the canonical GT-dinucleotide at the 5′-splice site of intron 9, we experimentally assessed the consequences of the deletion on the transcript level. Primers located in exons 7 and 11 of the *MIA3* gene were used to amplify cDNA from a DSRA affected and a control dog. The DSRA affected dog expressed a transcript that lacked 111 nucleotides consisting of the entire length of exon 8 and 9, confirming the existence of an aberrantly spliced *MIA3* transcript in the affected dog, XM_005640835.3:r.3712_3822del (Figure 4). The identity of the RT-PCR band was confirmed by direct Sanger sequencing. This exon skipping on the mRNA level is predicted to result in a deletion of 37 amino acids from the wild type *MIA3* protein, XP_005640892.1:p.(Val1238_Lys1274del).

## 4. Discussion

Hereditary dental pathologies are rare in dogs. While cases of combined skeletal and ocular anomalies have been reported in several dog breeds [3,4,5,6,7,14], according to our knowledge, a hereditary syndrome comprising brittle, translucent and discolored teeth, disproportionate growth and progressive vision loss, has so far not been described in dogs. Further studies to characterize the phenotype of DSRA affected dogs in detail are currently performed and will be published separately.

We were able to delineate the DSRA locus to a 5.8 Mb critical interval. Whole genome sequencing identified a homozygous private splice region variant, *MIA3*:c.3822+3_3822+4del, that was confirmed to cause the skipping of two exons from *MIA3* transcripts. Genotypes at this variant co-segregated with the phenotype in a small family and were perfectly associated within a cohort of 18 cases and 22 controls.

Variants affecting the third and/or fourth nucleotide of an intron may affect the correct splicing of mRNA transcripts. Other known examples of functional variants at position +3 or +4 in 5′-splice sites include the *MKLN1*:c.400+3A>C variant in dogs with lethal acrodermatitis [15] or the *MBTPS2*:c.1437+4C>T variant in horses with the brindle 1 phenotype [16].

The *MIA3* gene encodes a transmembrane protein termed MIA SH3 domain ER export factor 3, which is also known under the alias name “transport and Golgi organization 1” (TANGO1) [17]. MIA3 plays an important role in the transport of secretory cargo including collagens from the endoplasmic reticulum (ER) to the ER-Golgi intermediate compartment [17,18,19,20]. Collagens represent abundantly secreted molecules in mammals and are needed throughout the whole body for bone mineralization, skin and tissue assembly [20]. Newly synthesized procollagens assemble within the ER lumen into rigid, rod-like triple helices that reach up to 450 nm in length and are too large for export by the conventional coat protein complex II coated vesicles of 60–90 nm diameter. MIA3 is involved in the organization of ER exit sites and specifically required as a component of large transport vesicles with bulky cargoes such as collagens [17,20,21].

In *Mia3^−^*^/*−*^ knockout mice, a complete deficiency of Mia3 leads to massive aberrations in the secretion of several collagens with almost complete lack of bone formation resulting in perinatal lethality. Heterozygous *Mia3^+^*^/*−*^ mice did not show any obvious phenotypic changes compared to wildtype mice [22].

A complete loss of MIA3 was also reported in an aborted human fetus homozygous for a p.Leu924Serfs* frameshift variant. This fetus had extremely thin bones resembling a lethal osteogenesis imperfecta phenotype [23]. Near-complete partial loss of MIA3 in human patients homozygous for a variant that caused partial skipping of exon 8 resulted in a syndromic disease with striking phenotypic similarities to the dogs of our investigation. Four affected children of a consanguineous family with normal parents had severe dentinogenesis imperfecta, short stature and various other skeletal abnormalities, insulin-dependent diabetes mellitus, sensineural hearing loss and mild intellectual disability. Two of the four affected children also had a mild retinopathy [24,25]. In the human patients, partial skipping of exon 8 was caused by a SNV, c.3621A>G, located in an exonic splice enhancer motif in the middle of exon 8 [25]. This suggests that the short exon 8, which spans only 22 bp, may be particularly sensitive to splicing defects.

The data from human patients and knockout mice suggest that a complete loss of MIA3 function probably cause lethal phenotypes in mammals. However, the phenotype of the human patients with partial skipping of exon 8 shows many parallels and comparable severity to the DSRA phenotype in Cane Corso dogs. We therefore speculate that the DSRA affected dogs retain some residual MIA3 activity. One possible explanation for such a residual activity could be the generation of a very small proportion of correctly spliced wildtype transcripts from the mutant allele. As the shorter mutant transcript is preferentially amplified during RT-PCR, a small proportion of wildtype transcripts might have escaped detection in our experiment (Figure 4A). Alternatively, as the skipping of exons 8 and 9 in DSRA affected dogs preserves the open reading frame, it is also possible that a mutant MIA3 protein with some residual function is expressed. The predicted mutant protein in DSRA affected dogs lacks 37 amino acids corresponding to amino acids 1204–1240 of the cytoplasmic coiled-coil 1 domain in the orthologous human protein (NP_940953.2) [25]. As the two transmembrane domains should be intact in the mutant protein, it is conceivable that this protein is expressed with the correct topology in the ER membrane and retains partial functional activity.

While the proposed MIA3 deficiency is likely to affect the export of many proteins, the clinical phenotype in DSRA partially recapitulates what is seen in dogs with other collagen defects. A lack of mature collagen type I leads to osteogenesis imperfecta that includes dentinogenesis imperfecta with brittle translucent and pink teeth [26,27,28,29,30]. In oculoskeletal dysplasia (OSD) a lack of collagen type IX is responsible for the phenotype [6]. We speculate that in DSRA affected dogs the production and secretion of all types of collagen is impaired to some degree.

## 5. Conclusions

We describe a new syndrome tentatively termed dental-skeletal-retinal anomaly (DSRA) with autosomal recessive inheritance in Cane Corso dogs. A genomic *MIA3* splice region variant leading to aberrant splicing represents a compelling candidate causative variant for DSRA. Our findings enable genetic testing in Cane Corso dogs, which can be used to detect unaffected carriers and avoid the unintentional breeding of further affected puppies. The studied dogs might serve as animal model to further elucidate the function of MIA3 in mammals.

## Figures and Tables

**Figure 1 genes-12-01497-f001:**
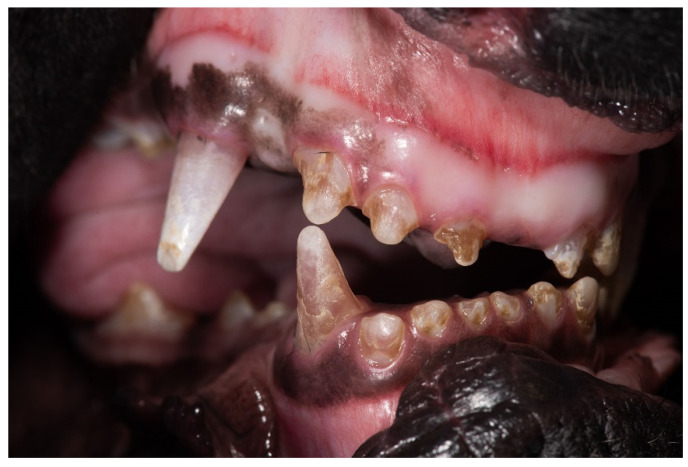
Oral cavity of a DSRA affected Cane Corso dog. All teeth appear translucent and show marked brown discoloration with multifocal enamel defects. Used with permission Utrecht University.

**Figure 2 genes-12-01497-f002:**
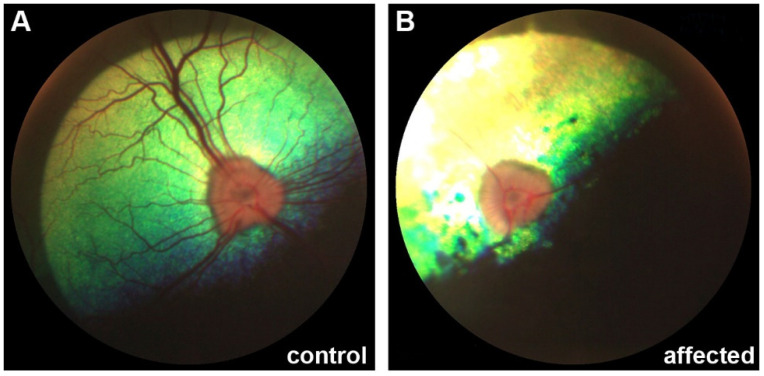
Ophthalmological phenotype of DSRA affected dogs. (**A**) Retina of a control Cane Corso; (**B**) Retina of a 5-months-old DSRA affected Cane Corso displaying signs of progressive retinal atrophy (PRA).

**Figure 3 genes-12-01497-f003:**
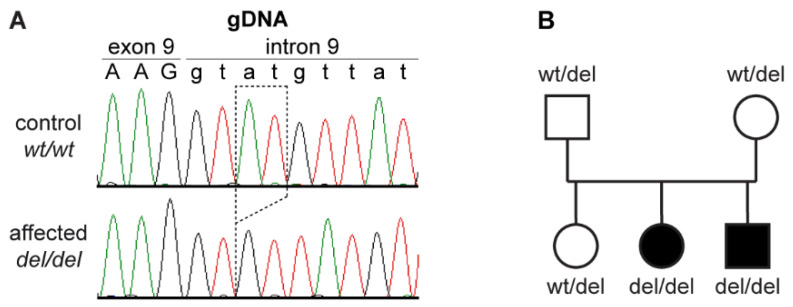
Details of the *MIA3*:3822+3_3822+4del variant. (**A**) Sanger sequencing electropherograms of a control and a DSRA affected dog illustrate the 2 bp deletion at the beginning of intron 9 of the *MIA3* gene; (**B**) Pedigree of a Cane Corso family shows the expected co-segregation of the genotypes at the deletion with the DSRA phenotype assuming monogenic autosomal recessive inheritance.

**Figure 4 genes-12-01497-f004:**
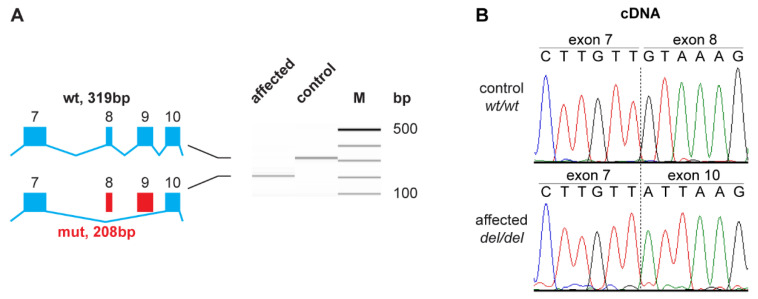
Experimental confirmation of the splice defect on the transcript level. (**A**) RT-PCR analysis of *MIA3* transcripts in an affected and a non-affected dog. Fragment Analyzer bands of the products in the control animal show the expected 319 bp band, while only a shorter 208 bp band is visible in the affected animal. The difference in fragment sizes is 111 bp, which corresponds to the combined length of exons 8 and 9; (**B**) Sanger sequencing of the RT-PCR products confirmed the breakpoints of the aberrant splicing product.

**Table 1 genes-12-01497-t001:** Results of variant filtering in a DSRA affected dog against 789 control genomes.

Filtering Step	Homozygous Variants
All variants	2,286,318
Private variants	1776
With SnpEff impact high, moderate or low	8
In critical intervals	1

**Table 2 genes-12-01497-t002:** Association of the genotypes at the *MIA3*:3822+3_3822+4del variant with DSRA in 40 Cane Corso dogs.

Phenotype	*wt*/*wt*	*wt*/*del*	*del*/*del*
DSRA cases (*n* = 18)	-	-	18
Non-affected control dogs (*n* = 17)	8	9	-
Dogs with unknown phenotype (*n* = 5)	5	-	-

## Data Availability

The accessions for the sequence data reported in this study are listed in Appendix A.

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
