# Peer review of "MIA3 Splice Defect in Cane Corso Dogs with Dental-Skeletal-Retinal Anomaly (DSRA)"

_genes, 2021, doi:10.3390/genes12101497_

Round 1

Reviewer 1 Report

Overall, the manuscript is well written and the results are simple and enough to support the conclusions. These results are important for a better understanding of DSRA in dogs. I would like to point out some minor comments above.

p3L107-109 please add PCR reaction information (like L121-123).

The authors investigated a heritable syndrome DSRA in Cane Corso dogs and found a deletion in the MIA3 gene caused the phenotype. This is an autosomal recessive disorder, and related phenotypes were found in humans and mice with MIA3 gene mutation/deletion.
In Fig.4A, please show the real gel-electrophoresis picture (not diagram).
Why is this MIA3 deletion specific in Cane Corso dogs? Is this deletion inherited from one particular ancestor dog? Because all mutations found in this study were the same, it seems that a mutation occurred in a specific dog at some point in the past.

Author Response

(1)

Overall, the manuscript is well written and the results are simple and enough to support the conclusions. These results are important for a better understanding of DSRA in dogs. I would like to point out some minor comments above.

p3L107-109 please add PCR reaction information (like L121-123).

Response: We added further PCR details as requested.

(2)

The authors investigated a heritable syndrome DSRA in Cane Corso dogs and found a deletion in the MIA3 gene caused the phenotype. This is an autosomal recessive disorder, and related phenotypes were found in humans and mice with MIA3 gene mutation/deletion. In Fig.4A, please show the real gel-electrophoresis picture (not diagram).

Response: We are not sure that we understand the comment from the reviewer. Figure 4A shows a schematic representation of the wildtype and mutant splicing pattern on the left side. The right half of Figure 4A shows the “real gel-eletrophoresis picture” and the experimental data! Please note that we performed fragment size analysis of PCR products by capillary gel electrophoresis on a 5200 Fragment Analyzer instrument (see Methods, chapter 2.8, lines 128-129). Bands on this instrument look much “nicer” than bands on traditional agarose gels. We did not change the manuscript with respect to this comment.

(3)

Why is this MIA3 deletion specific in Cane Corso dogs? Is this deletion inherited from one particular ancestor dog? Because all mutations found in this study were the same, it seems that a mutation occurred in a specific dog at some point in the past.

Response: We agree with the reviewer that the causal MIA3 deletion happened in a single founder dog that must have been an ancestor to all affected dogs in this study. This is a very typical situation in purebred dogs that are bred in strictly closed populations with a fairly high degree of inbreeding. We speculate that the initial mutation event may have happened ~5-10 generations ago. However, as we don’t have access to samples from the ancestors, we cannot experimentally trace the origin of the pathogenic allele. We did not change the manuscript with respect to this comment.

Reviewer 2 Report

The manuscript entitled “MIA3 Splice Defect in Cane Corso dogs with Dental-Skeletal-2 Retinal Anomaly (DSRA)” describes a novel hereditary syndrome, dental-skeletal-retinal anomaly (DSRA) in Cane Corso dogs. The authors identified a novel mutation in the MIA3 gene in affected dogs by using an elegant genomic approach. I enjoyed reading the manuscript and would strongly recommend the publication of this manuscript with minor suggestions.

The exon skipping by the identified mutation is intriguing. 

  1. I would suggest performing RT-PCR of the MIA3 gene in other tissues including muscle in a control dog(s) to make sure that the exon skipping does not represent just a splicing isoform. 
  2. It would be helpful to add examples of exon skipping phenomenon caused by mutations that do not directly affect the canonical GT-donor or AG-acceptor nucleotides in Discussion.  

Author Response

(1)

The manuscript entitled “MIA3 Splice Defect in Cane Corso dogs with Dental-Skeletal-2 Retinal Anomaly (DSRA)” describes a novel hereditary syndrome, dental-skeletal-retinal anomaly (DSRA) in Cane Corso dogs. The authors identified a novel mutation in the MIA3 gene in affected dogs by using an elegant genomic approach. I enjoyed reading the manuscript and would strongly recommend the publication of this manuscript with minor suggestions.

Response: Thank you for the compliments.

(2)

The exon skipping by the identified mutation is intriguing.

I would suggest performing RT-PCR of the MIA3 gene in other tissues including muscle in a control dog(s) to make sure that the exon skipping does not represent just a splicing isoform.

Response: We actually performed the RT-PCR in three different tissues from the affected dog (muscle, liver, skin). In all three tissues, the amplicon representing the aberrant transcript with the two missing exons was formed and no wildtype transcript was amplified. We decided to show only the band from skeletal muscle in Figure 4A as this was the strongest band and most comparable in intensity to the control. Unfortunately, we currently do not have suitable other tissue samples from healthy dogs and cannot perform the requested additional experiment. However, as the difference is also present between cDNA from skin samples of the affected versus the control dog, tissue-specific alternative splicing is at least excluded in skin.

(3)

It would be helpful to add examples of exon skipping phenomenon caused by mutations that do not directly affect the canonical GT-donor or AG-acceptor nucleotides in Discussion.

Response: We expanded the discussion on the MIA3 splicing defect in human patients, which is due to an exonic SNV that also does not affect the canonical AG/GT consensus dinucleotides (lines 237-240). We also added a small paragraph with two additional examples of variants at the +3 and +4 position of the intronic 5’-splice site that cause splicing defects in other genes (lines 208-212).